# Intelligent Insulin vs. Artificial Intelligence for Type 1 Diabetes: Will the Real Winner Please Stand Up?

**DOI:** 10.3390/ijms241713139

**Published:** 2023-08-24

**Authors:** Valentina Maria Cambuli, Marco Giorgio Baroni

**Affiliations:** 1Diabetology and Metabolic Diseaseas, San Michele Hospital, ARNAS Giuseppe Brotzu, 09121 Cagliari, Italy; vale.m.cambuli@gmail.com; 2Department of Clinical Medicine, Public Health, Life and Environmental Sciences, University of L’Aquila, 67100 L’Aquila, Italy; 3Neuroendocrinology and Metabolic Diseases, IRCCS Neuromed, 86077 Pozzilli, Italy

**Keywords:** intelligent insulin, artificial intelligence (AI), glucose-responsive insulin (GRI), concanavalin A (ConA), phenylboronic acid (PBA), glucose oxidase (GOx), automated insulin delivery systems, continuous glucose monitoring (CGM), machine learning (ML), deep learning (DL)

## Abstract

Research in the treatment of type 1 diabetes has been addressed into two main areas: the development of “intelligent insulins” capable of auto-regulating their own levels according to glucose concentrations, or the exploitation of artificial intelligence (AI) and its learning capacity, to provide decision support systems to improve automated insulin therapy. This review aims to provide a synthetic overview of the current state of these two research areas, providing an outline of the latest development in the search for “intelligent insulins,” and the results of new and promising advances in the use of artificial intelligence to regulate automated insulin infusion and glucose control. The future of insulin treatment in type 1 diabetes appears promising with AI, with research nearly reaching the possibility of finally having a “closed-loop” artificial pancreas.

## 1. Introduction

In the treatment of type 1 diabetes, the importance of the relationship between the intensity of insulin treatment, glycaemic control, and the reduction of complications was clarified by the DCCT study [1]. The current treatment of type 1 diabetes aims to achieve all these objectives, and from the point of view of glycaemic control and reduction of complications, the results appear certainly positive [2].

At the present time, multiple daily injections (MDI) of insulin are the current standard treatment for type 1 diabetes. Advances in insulin therapies, including the development of insulin analogues with a faster or prolonged time of action, novel or targeted approaches to administration, new glucose monitoring systems, and novel pens and devices for administration have all contributed to the improved treatment of T1D patients. Despite these developments, many patients do not attain the glycaemic targets set by international guidelines [3]. Data from the Type 1 Diabetes Exchange clinic registry showed that the average A1C for 16,057 participants with T1D was 8.4% [4]. Moreover, the life expectancy of patients who develop T1D at an age before 10 years is still 10–16 [5] years less than non-diabetic population.

There are certainly open questions that current therapeutic tools, newer insulin, and glucose monitoring are still unable to resolve. In particular, the frequency of hypoglycaemia, weight gain, and glycaemic variability remain difficult to control. The DCCT itself demonstrated how the intensity of insulin treatment was accompanied by an increase in the frequency and severity of hypoglycaemia [6], and at the same time, by a progressive increase in weight in the intensively treated group [7].

Thus, several unmet needs are still present in the current treatment of type 1 diabetes. Together with hypoglycaemia and weight gain, glucose variability [8] and unsatisfactory glucose control [4] are very important open concerns in type 1 diabetes treatment. Finally, compliance (missed/delayed injection) is another issue that needs addressing.

Research in type 1 diabetes treatment has been addressed to solve the above-mentioned unmet needs. To these aims, the future of insulin therapy is directed toward two main areas: (1) the development of “intelligent insulins,” capable of auto-regulating their own levels according to glucose concentrations, or (2) the use and progress of artificial intelligence, exploiting its own learning capacity to provide decision support systems to further improve the treatment of diabetes.

This review will discuss the most recent advances in the two areas of development of insulin therapy, providing updated insights into the progress on intelligent insulins and artificial intelligence, the latter capable of displaying human capabilities such as reasoning, learning, planning, and analysing the effects of previous actions with independent adaptation.

## 2. Intelligent Insulins

The concept of intelligent/smart insulins is straightforward, and it can be summed up as “from complexity to simplicity.” In essence, it is a question of creating an insulin capable of self-regulating according to changes in blood sugar, so-called “glucose responsive insulin” (GRI), eliminating the need to consider all the variables involved in dose calculation (calories, carbohydrates, exercise, time, etc.).

GRI may be formulated as polymer-based systems, wherein insulin is encapsulated within a glucose-responsive polymeric matrix-based vesicle or hydrogel [9] or as a molecular GRI analogue system, which involves the introduction of a glucose-sensitive motif to the insulin molecule; in either case, its formulation confers glucose-responsive changes to insulin bioavailability or hormonal activity [10,11].

Polymer-based technologies employ the sequestration of insulin within a matrix suitable for subcutaneous injections. Stimuli-responsive polymers (hydrogels) can convert environmental stimuli (i.e., temperature, pH, ionic strength, glucose concentration) into the signal to trigger the change in physical properties of the polymers themselves [12,13]. The matrix, in principle, senses the glucose concentration and releases a proportional amount of insulin.

To date, the principal mechanisms underlying artificial smart GRI delivery systems are (Figure 1):Glucose-binding proteins, a class that includes lectins, like concanavalin A (ConA);Phenylboronic acid (PBA);Glucose oxidase (GOx).

### 2.1. Glucose-Binding Proteins

In 1979, a stable, biologically active glycosylated insulin derivative that was complementary to the major combining site of concanavalin A (ConA) was synthesized as a pioneering GRI system. ConA is a protein of the lectin family that can reversibly and specifically bind to glucose and mannose with a high affinity [14].

Hormone release was proportional to the quantity of glucose present [15], and the complex was designed to sequester insulin in the subcutaneous space during normoglycaemia and release the hormone during hyperglycaemia via competition with ambient glucose molecules.

Responsive hydrogel(s) were formulated from glucosyloxyethy1 methacrylate (GEMA), N,N′-methylene-bis-acrylamide (MBAAm) [16,17] and porous poly(hydroxyethyl methacrylate) (PHEMA) [18]. The rise of glucose concentration led to a decrease of gel density, with an insulin release.

It is also possible to conjugate insulin to various glucose-like molecules: the combination of glucose-modified insulin and the glucose-binding lectin, ConA, leads to a dissociation/reassociation under increased/decreased glucose concentrations [14,19].

ConA’s immunogenicity and mitogenicity limit the clinical translation [20] of these approaches, and a major challenge is that the self-regulation of insulin release from the GRI-ConA system in response to different glucose levels will not be always reproducible [14].

### 2.2. Phenylboronic Acid (PBA)

Phenylboronic acid (PBA) is a diol-binding element that exhibits a strong affinity for sugars, such as D-glucose [21]. The competitive binding of the hydroxyl groups of D-glucose to borate ion increases the swelling of the glucose-PBA complex. One major concern for practical application is that the pH dependence of PBA ionization at physiological conditions is not guaranteed. Various approaches have been conducted to use PBA in GRI systems.

As GRI, it was demonstrated that a modified insulin, carrying a PBA and a polyol group attached to the LysB29 sidechain, forms a high molecular weight multimeric complex that dissociates as D-glucose concentration rises [10]. A similar mechanism was applied to insulin degludec [22] and detemir [23]. Insulin detemir was modified through the incorporation of an aliphatic domain to facilitate hydrophobic interactions and a PBA for glucose sensing. This GRI affords long-lasting and glucose-responsive activity in the diabetic mouse, providing a potentially improved therapeutic strategy for diabetes management [23].

PBA’s reversible binding to glucose provides a potential mechanism for designing insulin formulations that respond to changes in blood glucose concentrations, so various approaches (glucose-triggered swelling, dissociation, competitive replacement) have been conducted to use PBA in GRI systems [12]. Several matrices (gels, micelles, capsules) have been synthesized with the use of PBA. Thanks to PBA’s ability to switch from a hydrophobic form to a hydrophilic form, insulin may or may not be released in response to glucose levels [12].

Various hydrogels were prepared with in vitro and, in some cases, in vivo results. Matsumoto described a catheter-combined device using PBA gel that was suitably scaled for mouse model experiments and demonstrated that it could serve as an “artificial pancreas” in diabetic mice [24]. The effect of the device on glucose metabolism in streptozotocin (STZ)-induced type 1 diabetes mice, was studied: seven days after implantation of the recombinant human insulin-loaded device, there was a marked reduction in glycaemia and water intake, with at least 3-week durability, potentially dependent of the volume of the reservoir [24].

### 2.3. Glucose Oxidase (GOx)

Contrary to ConA and PBA-based systems, glucose oxidase (GOx)-coupled systems do not directly bind or form a complex with glucose molecules to exhibit sensitivity. Utilizing GOx enzymes, circulating glucose and oxygen can be initially oxidized into gluconic acid and hydrogen peroxide end-products. The gluconic acid is used as a trigger for responsiveness when coupled with pH-responsive materials in the second step of this process. Thus, entrapping or cross-linking GOx with pH-responsive polymers can cause rapid-phase swelling/deswelling in response to environmental glucose.

Glucose oxidase (β-D-glucose:oxygen 1-oxidoreductase, EC 1.1.3.4) is a flavoprotein that catalyses the oxidation of β-D-glucose at its first hydroxyl group, utilizing molecular oxygen as the electron acceptor, to produce D-glucono-delta-lactone (gluconic acid) and hydrogen peroxide (H_2_O_2_) [25]. The glucose-oxidation reaction catalysed by GOx changes the physiological environment, modifying pH, H_2_O_2_ concentration, and O2 levels.

A polymer with tertiary amino groups is protonated by a decrease in the pH value of a medium, and the hydrophilicity of the polymer increases. A glucose-responsive polymer membrane can be obtained by combining a poly amine membrane and GOx immobilized membrane since the pH value of the medium will decrease by the formation of gluconic acid produced from the reaction of GOx with glucose. A GRI system could be prepared from N, N-diethylaminoethyl methacrylate [26], poly methacrylic acid-g-ethylene glycol [27], acid-responsive chitosan matrix, or GOx/catalase nanocapsules [28]. A glucose-responsive microgel composed of an acid-responsive chitosan matrix, GOx/catalase (CAT) nanocapsules, and recombinant human insulin demonstrated, in six streptozotocin-induced type 1 diabetic mice, a prolonged self-regulated profile of insulin release as a function of glucose concentration for encapsulated microgel [28]. Another in vivo microdevice tested in rats, GRI consisted of an albumin-based bioinorganic membrane that utilizes GOx/CAT and manganese dioxide nanoparticles: during in vitro testing, the microdevices showed glucose-responsive insulin release over multiple cycles at clinically relevant glucose concentrations. In vivo, the microdevices were able to counter hyperglycaemia in diabetic rats over a one-week period [29].

An alternative strategy of achieving a pH responsive release of insulin is to encapsulate insulin inside an acid-disintegrable formulation, such as hydrogels, liposomes, and polymeric vesicles [12,30]. A positively charged acid-disintegrable microgel loaded with insulin by electrostatic interactions and covalently immobilized with GOx and catalase by inverse emulsion polymerization was formulated, aiming for a glucose-regulated insulin release by utilizing GOx/catalase cascade enzymatic reactions to trigger local pH decrease. A local pH decrease within the microgels leads to the diminishment of the net surface negative charges of encapsulated insulin, with a consequent insulin release at high glucose levels [30]. Using chemically modified dextran as an acid-degradable and biocompatible matrix material, a nano-network for glucose-regulated insulin delivery was prepared. The nanoparticles used to form this nano-network comprised four components: an acid-degradable polymeric matrix, polyelectrolyte-based surface coatings, encapsulated glucose-specific enzymes (GOx and catalase), and recombinant insulin. Dextran was selected due to its biocompatibility and biodegradability. The degradation of the nano-network with insulin release was glucose-mediated (facilitated in hyperglycaemia and inhibited in normal glucose levels) in both in vitro and in vivo studies. The blood glucose levels of diabetic mice treated with one injection of nano-network were stably maintained in the normoglycaemia (<200 mg/dL) range for up to 10 days without peaks of hyperglycaemic or hypoglycaemic states [31].

Thus, compared with ConA and PBA systems, glucose-oxidase coupled systems have shown the most promise in vivo in mice; other studies have shown glucose-sensitive insulin release in animal models, with a duration of efficacy ranging from 24 h to 14 days [32,33,34,35].

### 2.4. Glucose-Responsive Microneedle Patches for Diabetes Treatment

Several glucose-responsive microneedle (MN)-array patches have been developed to treat diabetes. Previously, open-loop MN-arrays [36,37] were described, but only a closed-loop MN-array could secrete a certain amount of insulin proportional to glucose concentration [38]. In practice, a GRI-MN-array could represent the ideal approach for smart insulin delivery.

Yu et al. [39] reported a GRI delivery device using a painless MN-array patch (“smart insulin patch”) containing glucose-responsive vesicles which were loaded with insulin and GOx enzymes. The vesicles were self-assembled from hypoxia-sensitive hyaluronic acid conjugated with 2-nitroimidazole, a hydrophobic component that can be converted to hydrophilic 2-aminoimidazoles through bioreduction under hypoxic conditions. The local hypoxic microenvironment caused by the enzymatic oxidation of glucose in the hyperglycaemic state promoted the reduction of hypoxia-sensitive hyaluronic acid, which rapidly triggered the dissociation of vesicles and subsequent release of insulin. Vesicles were loaded into a MN-array patch and disassembled in case of interstitial hyperglycaemia with insulin delivery. The smart insulin patch effectively regulated the blood glucose in (STZ)-induced diabetic mice [39].

More recently, several types of MN-array patches have been described. For example, a dissolvable MN patch was fabricated using pullulan, a water-soluble polysaccharide with excellent film-forming ability. After application into human abdominal skin in vitro, the microneedles dissolved within 2 h, releasing up to 87% of insulin. When stored at 4 °C, 20 °C, and 40 °C for 4 weeks, insulin was able to retain its secondary structure, and these MNs were non-cytotoxic toward human keratinocytes [40]. Another dissolving and glucose-responsive insulin-releasing MN patch system was formulated with dissolving and biodegradable gelatine and starch materials, which encapsulated glucose-responsive insulin-releasing gold nanocluster nanocarriers. The gold nanocluster nanocarrier drugs enabled the MN patches to obtain glucose-responsive insulin-releasing behaviour. In in vivo studies, one transdermal application of this patch regulated glycaemia in type 1 diabetic mice in the euglycemic range for 24–48 h [41]. Also, PBA was employed for preparing a MN patch: a purified polymer-bearing pendant PBA motif was combined with a multivalent diol crosslinker to prepare dynamic-covalent hydrogel networks that could be loaded into an MN-array. Insulin release from these materials was accelerated in the presence of glucose, and this activity was confirmed in a diabetic rat model [42] Moreover, short-term blood glucose control in a diabetic rat model following the application of the device to the skin confirms insulin activity and bioavailability. Accordingly, dynamic-covalent crosslinking facilitates a route for fabricating microneedle arrays circumventing the toxicity concerns of in situ polymerization, offering a convenient device form factor for therapeutic insulin delivery [42].

So far, one of the major concerns for the clinical application of MN-array patches is the complexity of the manufacturing process. Very recently, three-dimensional printing technology was employed for the fast fabrication of MN patches. This technique can precisely print microneedles in a few seconds. Thus, 3D-constructed MN patches can efficiently deliver insulin into diabetic mice’s skin by injection, resulting in the effective control of blood glucose levels [43].

In summary, the most studied approaches for developing glucose-responsive insulin delivery have been those based on glucose oxidase, owing to its high specificity for glucose, its current usage in glucose sensors, and the wide array of pH-responsive materials. However, the enzymatic conversion of glucose remains unreliable and slow, and the release of insulin from these nanoparticles is inversely related to glucose concentrations. Glucose binding proteins provide high specificity and binding to glucose; however, limited progress has been made toward eliminating foreign body immune responses. Although small-molecule binders lack glucose specificity, new approaches such as multiplexing PBAs are being explored to address these limitations.

### 2.5. Clinical Use of GRI Systems 

At present, there is no clinical application for smart insulins. In 2010, Merck & Co., Inc. acquired a smart insulin called MK-2640. MK-2640 is an insulin analogue able to bind the lectin receptor mannose receptor C-type 1 (MRC1). The competitive binding between MK-2640 and glucose to MRC1 was exploited to tune the blood clearance rate of MK-2640 [44,45]. Despite promising results in studies with minipigs and dogs [46], results in humans have failed, leading to the discontinuation of the trial. Compared to pre-clinical studies, MK-2640 did not display a glucose-dependent change in MK-2640 systemic clearance in T1D subjects between euglycemic and hyperglycaemic conditions [47].

## 3. Artificial Intelligence: General Concepts

Artificial intelligence (AI) focuses on how computers learn from data and mimic human thought processes. AI increases its learning capacity, provides decision support systems that have the potential to transform the future of healthcare [48], and is extensively employed in diabetes care to further improve the treatment of diabetes. Both machine learning (ML) and deep learning (DL), with their wide spectrum of increasingly complex algorithms, are used in different diabetes fields, including prediction, diagnosis, glucose management, screening of complication, and others [49,50] (Figure 2).

The most common type of AI methodologies in routine clinical use is expert systems (ES). ES are systems with the ability to capture expert knowledge, facts, and reasoning techniques to help care in routine work. ES attempt to mimic the clinician’s expertise by applying inference methods to help in decision support or problem-solving [49]. The most used ES methods in diabetes are (1) rule-based reasoning (RBR), in which the computer is capable of finding solutions to problems that should be solved by an expert; (2) case-based reasoning (CBR), an AI technique which solves new problems based on the solutions of similar past problems [51] (for example, using a database of previously solved problems, throughout four steps (Retrieve, Reuse, Revise, Retain), it confirms, improves, and integrates the solutions to be reused in the future, if they work); (3) fuzzy systems, which present an intricate logic with infinite intermediate states between zero and one, allowing computers to better represent the complexity of reality [49].

Machine learning algorithms are characterized by the ability to learn over time without being explicitly programmed [49]. Systems can learn knowledge and patterns automatically from experience or existing data without being explicitly instructed [52]. Experts typically ‘‘train’’ AI systems with large amounts of data and algorithms, which enable the machine to examine relationships and learn from them [50]. Methods in ML include decision trees, artificial neural networks (ANN), genetic algorithms, or support vector machines. A clear example of a successful application can be found in the field of diabetes, where insulin subcutaneous infusion in an artificial pancreas system is adapted learning from continuous glucose monitoring in patients affected by type 1 diabetes [49]. An ANN simulates the neuron behaviours by algorithms, comprehending three layers of nodes: input, hidden, and output [52].

Deep learning is, formally, the more evolved branch of ML that could be considered an implementation of artificial neural networks. Adding more hidden layers, deep learning extends the ANN structure to deep neural networks (DNN) for better generalization, which extracts data features and learns representations with thousands or even millions of parameters [53]. To date, there are five popular DNN architectures for diabetes: deep multilayer perceptron (a machine learning algorithm used for supervised learning of various binary classifiers), convolutional neural networks, recurrent neural networks, autoencoders, and restricted Boltzmann machines [52]. Deep learning algorithms can be divided into supervised and unsupervised learning. Compared to supervised learning, unsupervised algorithms do not require predefined supervised inputs for learning [52].

### 3.1. Artificial Intelligence in Glucose Management

The main challenge in the treatment of type 1 diabetes is the complete automation of blood glucose control coupled with the automatic infusion of insulin, alone or together with glucagon. This is the so-called “artificial pancreas” (AP), or the technological replacement of the pancreatic beta cell destroyed by the autoimmune process. More properly, at the current state of technological advances, it is more correct to define the artificial pancreas as an automated insulin delivery (AID) system, considering that only hybrid AIDs are currently available; hybrid AIDs are insulin pump systems that automatically increase or decrease basal insulin delivery in response to sensor glucose values but also have the capacity to deliver automatic correction boluses. However, the user still needs to dose prandial insulin manually [54].

These systems consist of three components: a continuous glucose monitoring (CGM) through an interstitial sensor, a subcutaneous insulin infusion pump, and one or more algorithms that relate these two physical components so that they regulate, with the greatest possible precision, the patient’s blood glucose in the glucose targets defined for the algorithm itself [55,56]. The key element of the prototypes of AP is the algorithm and, to date, there are three algorithms used for this purpose: Proportional Integral Derivative (PID), Model Predictive Control (MPC), and Fuzzy Logic (FL) (Figure 3). 

The PID algorithm determines the insulin dose based on (1) the difference between the glycaemic target and the measured glucose (proportional component), (2) the rate of change of blood glucose (derived component), and (3) the difference between the areas underlying the curves described by target blood glucose and measured blood glucose, considering the insulin administered (integral component). The three components are then combined for the final calculation of the dose of insulin to be administered. Most PID algorithms also include continuous insulin on board (the amount of insulin still active due to its half-life) to reduce the risk of hypoglycaemia.

The MPC algorithm predicts glycaemic values and simultaneously adjusts the insulin infusion rate, considering the time required for the absorption of insulin administered subcutaneously, the insulin present (onboard), and daily and postprandial changes in blood glucose. The process is repeated every 5–15 min considering other available information, such as carbohydrate intake or exercise. This algorithm also has strategies to prevent inappropriate changes in blood glucose (both hypo and hyper). The algorithm working according to FL adjusts the insulin dose based on the measured blood glucose levels and the direction and speed of change in glycaemic values, imitating the logic followed by doctors treating diabetes [57,58].

### 3.2. Clinical Applications of Automated Insulin Delivery Systems

Recent years have seen exponential growth in the technology of automated insulin delivery systems with the aim of improving the time in range (TIR) until reaching and exceeding the objectives of good compensation [3,59] and glycosylated haemoglobin (HbA1c) [1,3].

It has been shown that, in adolescent and adult T1D patients, the TIR increased by 11 percentage points compared to the control group after 6 months of using a closed-loop system, with a clear decrease in the time in hypoglycaemia and with the improvement of HbA1c [60,61]. In this multicentric, randomized trial, the AID system consisted of a pump (t:slim X2 insulin pump with Control-IQ Technology, Tandem Diabetes Care) and a continuous glucose monitoring (Dexcom G6, Dexcom). This system is a third-generation descendant of DiAs, a mobile closed-loop system developed at the University of Virginia and subsequently implemented as inControl by TypeZero Technologies [62,63] consisting of a to-range adaptive MPC algorithm located on pump. The algorithm located in this system makes it possible to increase, decrease, block the pre-set basal insulin delivery in the machine by the clinician, and, in case of exceeding the maximum glycaemic target, deliver a corrective insulin bolus every hour.

The validity of a PID algorithm with insulin feedback with adaptive insulin limits, located on hybrid closed-loop pump 670G, Medtronic (pump plus continuous glucose monitoring Sensor G3), was probed in a six-month randomized controlled trial in adults with type 1 diabetes [64]: the TIR increased from 55% at baseline to 70% after 26 weeks and remaining unchanged at 55% in the control group. In AID group, HbA1c was significantly lower, and diabetes-specific positive well-being was higher. Encouraging results have also been obtained in paediatric-age patients [65]. The PID algorithm was then integrated with FL components, also allowing the delivery of a fully automated baseline and the automatic delivery of correction insulin boluses every 5 min, with excellent results in increasing TIR and decreasing time in hyperglycaemia [66,67] up to the most advanced model currently on the market, the Medtronic 780G, with Sensor G4 [68].

A treat-to-target adaptive MPC algorithm has been successfully used in the other hybrid machines currently approved for use in patients with type 1 diabetes: the Omnipod 5 (currently the only hybrid patch pump with a glucose target from 110 to 150 mg/dL) [69], the Diabeloop DBLG1 (algorithm on dedicated device and target 100–130 mg/dL) [70,71] and the Ypsopump mylife CamAPS FX (algorithm on smartphones, which is also approved in children over 1 year of age and during pregnancy for, respectively, particularly large and/or restricted targets allowed 80–198 mg/dL) [72,73]. All of these systems are integrated with Dexcom G6.

Evidence from numerous RCTs and real-world studies supports the safety and efficacy of the use of AID systems in young, school-aged paediatric populations and adolescent/adult populations [74]. In preschool children [75], adults aged 65 and older, and women with type 1 diabetes pre-existing to pregnancy, numerous studies do not yet allow a recommendation level A [74].

Currently, there is no clinical experience with fully automated systems, and, to our knowledge, there are no direct comparison studies between the different AID systems. The choice of one system or another is therefore dictated by the experience of the individual clinician and by the clinical characteristics of the patients, especially regarding glucose variability, working in shifts, mealtime, etc. Finally, it should be noted that the choice is often also linked to administrative organizations (national health systems, insurance, availability, etc.) that, in different countries, allow the prescription and/or refundability of the different AIDs described in our review.

### 3.3. The Future of Automated Insulin Delivery Systems

The two most important challenges for the future of automatic insulin delivery systems are represented by the use of an AID that is no longer hybrid, but fully automated also for the delivery of the bolus at meals, and by the so-called bi-hormonal systems where, in addition to insulin, glucagon or other hormones can be dispensed, with the main aim of avoiding hypoglycaemia.

Regarding fully automated AID systems [76,77], some evidence has emerged in small groups of patients. A randomized crossover trial was conducted in 11 adolescents aged 12–18 years with HbA1c > 7.5%, who missed one or more boluses in the past 6 months, to study the effectiveness of an automatic insulin delivery system integrated with a meal detection algorithm (MDA). During the period in MDA, participants experienced a decrease in the 4 h post-lunch incremental area under the curve, with no significant differences in TIR respect to AID without MDA [78].

Patients with T1D often have impairments in their counter-regulatory response to hypoglycaemia as well as insulin secretion, particularly if the duration of diabetes is long [79]. The currently usable AID systems prevent hypoglycaemia by decreasing, and even suspending, the delivery of basal insulin, but this modality has limitations, related, for example, to the half-life of the insulin already injected and/or the delay in glucose sensing. The first studies in hospitals on the bi-hormonal insulin/glucagon artificial pancreas were published in 2010 [80,81,82,83,84], with results somehow variable and discordant on TIR and TBR (time below range). One of the limitations of the bi-hormonal AID system is the half-life of glucagon and its instability at room temperature, requiring hormone replacement every 24 h. Recently, it was demonstrated [85] that dasiglucagon, a chemically stable glucagon analogue, can be used effectively in a prefilled dasiglucagon cartridge for 7 days.

On the clinical outcomes of bi-hormonal insulin/glucagon or insulin/pramlintide (synthetic analogue of amylin, a small peptide hormone that is circulated by pancreatic β cells together with insulin to counteract postprandial hyperglycaemic peaks [86]) AID, a recently published meta-analysis reviewed 17 studies involving a total of 438 patients, comparing bi-hormonal AID systems to insulin-only ones, as well as bi-hormonal systems using simple pump with or without predictive low-glucose suspension for hypoglycaemia [87]. The results of this meta-analysis suggest that bi-hormonal AID has a comparable effect on TIR compared with a single hormone but is associated with a longer time in the target range compared with a simple pump. The dual-hormonal system slightly reduced time in hypoglycaemia but increased the risk of gastrointestinal symptoms, both in insulin and glucagon systems and in an insulin and pramlintide system, and this adverse effect is a potential limitation in the use of a bi-hormonal system (Figure 4).

## 4. Conclusions

In this review, the authors wanted to discuss and compare the two principal technologies (basically, biotechnology and bioengineering) that will most likely lead to the technological replacement of pancreatic beta cell function, bringing patients affected by type 1 diabetes as close as possible to individuals with a normal tolerance to carbohydrates. 

Although the production of smart/intelligent insulins seems promising, in particular, with the use of GOx and in the form of microneedle devices [40,41,42,43], their use in humans appears difficult and far from clinical use at the moment. Ideally, the use of really “smart” or “intelligent” insulins might be the most acceptable for T1D patients, also considering the wearability of the GRI systems (they are almost invisible and not felt on the skin). Indeed, from our own personal experience, we understand that the expectations for an intelligent insulin without the need for glucose measurements are very high in T1D patients.

On the contrary, the use of artificial intelligence to integrate continuous glycaemic monitoring with automated insulin delivery, in formulations already commercially available, through hybrid pumps, is currently usable and increasingly widespread, safe, and effective [58,60,61,62,63,64,65,66,67,68,69,70,71,72,73,74,75,76,77]. It is also possible that the production of more concentrated insulins and the miniaturization of the systems may improve the wearability of AID systems. 

Both approaches are promising; however, at present, there is no doubt that it is artificial intelligence that can stand up as the most promising technology to achieve the most physiological replacement of insulin delivery in type 1 diabetes.

## Figures and Tables

**Figure 1 ijms-24-13139-f001:**
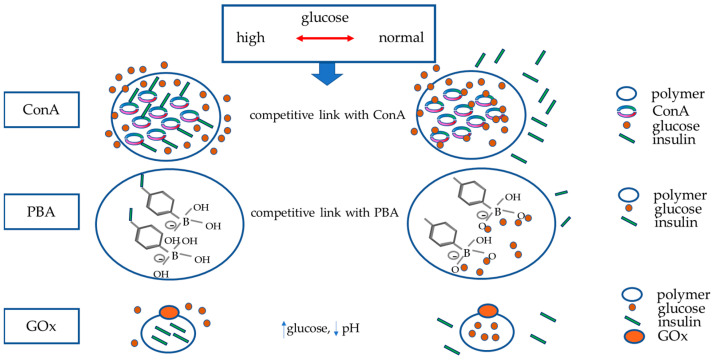
Principal mechanisms underlying artificial smart GRI delivery systems. The figure schematically represents the three mechanisms currently studied to build glucose-responsive insulin systems (nanoparticles or vesicles able to disassemble by swelling or degradation in response to increased glucose): Concanavalin A (ConA) and phenylboronic acid (PBA) release insulin once they are competitively bound by glucose. As blood sugar decreases, they will bind to insulin again. Polymers in which glucose oxidase (GOx) has been inserted will release insulin in response to the decrease in pH in the case of hyperglycaemia; GOx enzymatically converts glucose to gluconic acid, producing a drop in pH. This mechanism is also reversible when euglycemia is restored.

**Figure 2 ijms-24-13139-f002:**
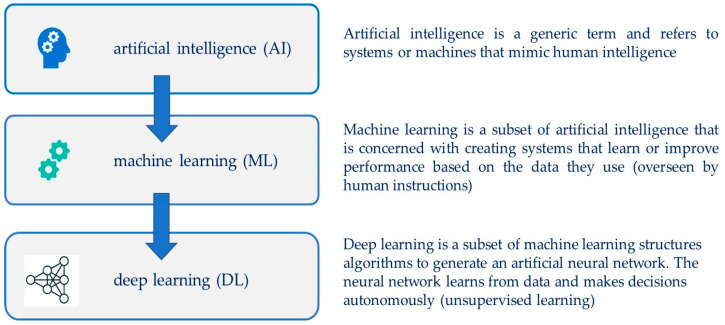
Artificial intelligence (AI).

**Figure 3 ijms-24-13139-f003:**
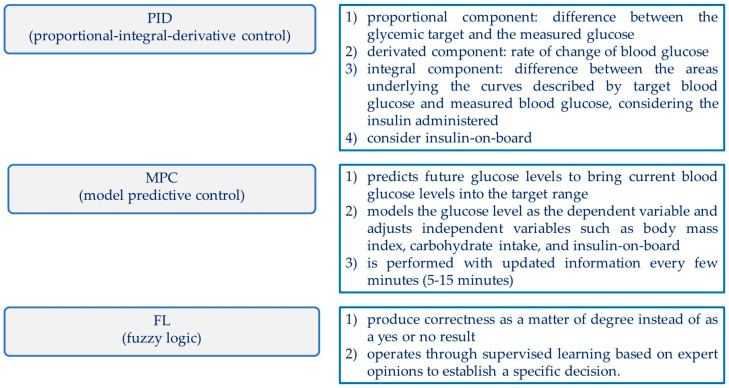
Algorithms used in automated insulin delivery systems [57,58].

**Figure 4 ijms-24-13139-f004:**
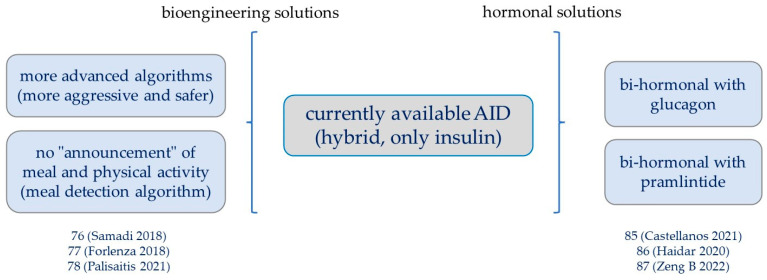
The future of automated insulin delivery systems (AID). Schematic representation of prospects for AID systems. Currently available AID systems are hybrids and use insulin alone. Algorithms that allow the full automation of the system are being tested or the use of other hormones, particularly glucagon [76,77,78,85,86,87].

## Data Availability

No new data were created or analyzed in this study. Data sharing is not applicable to this article.

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
