# Peer review of "Intelligent Insulin vs. Artificial Intelligence for Type 1 Diabetes: Will the Real Winner Please Stand Up?"

_ijms, 2023, doi:10.3390/ijms241713139_

Round 1

Reviewer 1 Report

This is a very comprehensive and well written review on a current topic on insulin control. Specifically, two different systems are discussed. A more research approach (e.g. not yet clinically relevant), "intelligent" insulin, and an already commercially available automated insulin delivery system. This reviewer has general comments.

Limit the introduction to the two glucose control systems. Discussions on unmet needs and compliance issues are not relevant. 

Please add reference after statement page 2/line 40. 

Author Response

Point-by-point response to Reviewer #1

Limit the introduction to the two glucose control systems. Discussions on unmet needs and compliance issues are not relevant. 

Please add reference after statement page 2/line 40. 

We thank the reviewer for the comments, and we agree that the discussion on unmet needs may not be necessary; we have therefore eliminated from the abstract the first sentence on unmet needs. However, we feel that for a non-specialist reader, these points are useful to understand the clinical issues involved, and we left a paragraph in the narrative of the introduction.

We have added a reference after the statement at page 1/line 40 (now pag. 1/line 36-38, underlined) on glucose targets according to the 2023 American Diabetes Association International Standard of Care, now reference n. 3 (underlined).

Reviewer 2 Report

Dear authors,
Thank you for this interesting update about intelligent insulins and automated insulin delivery.
This topic is obviously of concern, as controlling diabetes especially in patients with type 1 diabetes is really challenging.

First, concerning the form, I have only some comments:

-Line 52: there is a "t" missing at the end of "weigh"
-Line 74: I think there is a supplemental "s" at the end of "systems" that could be removed
-Line 209: the expression "stably maintained in the normoglycemia" is repeated

Now concerning the article itself, I think this article is well-written, clear and easy to read.

As a clinician, I would then be interested in having a little more details about the different systems of automated insulin delivery you mention.
Indeed, you well describe all the AID, which are quite different, but at the end, I do not really know the main differences between the systems in clinical practice.
Maybe you could describe a little more the main systems avalaible for now, and perhaps add what are the benefits and the drowbacks of each system.
I would also talk about the clinical situations that could incitate the clinician to use one system instead of another, because of the algorythm used in the system (especially in terms of meals, especially when patients do not announce meals, physical activity etc), if there are some studies available on this subject. Otherwise, this could be precised in the article.

Author Response

Point-by-point response to Reviewer #2

First, concerning the form, I have only some comments:

-Line 52: there is a "t" missing at the end of "weigh"

-Line 74: I think there is a supplemental "s" at the end of "systems" that could be removed

-Line 209: the expression "stably maintained in the normoglycemia" is repeated.

We apologize for the mistakes. We have corrected the points outlined and thoroughly checked and corrected all grammar throughout the text.

Now concerning the article itself, I think this article is well-written, clear and easy to read.As a clinician, I would then be interested in having a little more details about the different systems of automated insulin delivery you mention.

Indeed, you well describe all the AID, which are quite different, but at the end, I do not really know the main differences between the systems in clinical practice.

Maybe you could describe a little more the main systems avalaible for now, and perhaps add what are the benefits and the drowbacks of each system.

I would also talk about the clinical situations that could incitate the clinician to use one system instead of another, because of the algorythm used in the system (especially in terms of meals, especially when patients do not announce meals, physical activity etc), if there are some studies available on this subject. Otherwise, this could be precised in the article.

We thank the reviewer for his/her comments. Regarding the different systems of AI, we have added information on the available algorithms (at page 9, lines 376-379, 387-389, 392-397, underlined

Currently, there is no clinical experience with fully automated systems, and, to our knowledge, there are no direct comparison studies between the different AID systems. The choice of one system or another is therefore dictated by the experience of the individual clinician but also by the clinical characteristics of the patients, especially regarding glucose variability, working in shifts, mealtime, etc. Finally, it should be noted that often the choice is also linked to administrative organizations (national health systems, insurance, availability, etc.) that in different countries allow the prescription and/or refundability of the different AIDs described in our review. This is an important issue, and therefore this point has been discussed at page 9 lines 404-411, underlined.

Reviewer 3 Report

The review entitled “Intelligent insulin vs. artificial intelligence for type 1 diabetes: will the real winner please stand up?” by Drs Cambuli and Baroni offers a succinct and accurate description of the existing possibilities for the treatment of T1D. The comparative approach taken by the authors between intelligent insulins and AI devices is quite informative and well presented in the text; the four figures offer a good summary of the two approaches.

Although it is understandable that the normoglycemic control of type 1 diabetes is a major concern for the patients, it would be interesting if the authors could make a comment on how the patients themselves perceive the two possible treatments: the intelligent insulins or the artificial intelligence devices towards a near-final “close loop” artificial pancreas.

One minor remark concerns a suggestion for the consistent use of the initials for the disease in the text, according to the choice of the authors: T1D (see lines 38, 41, 42, 369) or T1DM (see lines 275, 416).

Author Response

Reviewer #3

Although it is understandable that the normoglycemic control of type 1 diabetes is a major concern for the patients, it would be interesting if the authors could make a comment on how the patients themselves perceive the two possible treatments: the intelligent insulins or the artificial intelligence devices towards a near-final “close loop” artificial pancreas.

The point risen by this reviewer is of particular importance. We have added a comment on this topic in the Conclusions (page 11, lines 467-472, and 476-478). We should point out, however, that this comment can only be speculative since there is no clinical experience regarding intelligent insulins, and it derives only from our clinical experience and from the experience, feedback, and expectations of the patients followed over many years by the authors.

One minor remark concerns a suggestion for the consistent use of the initials for the disease in the text, according to the choice of the authors: T1D (see lines 38, 41, 42, 369) or T1DM (see lines 275, 416).

We apologize for the inconsistent initials and changed all to T1D accordingly.